# Cross-Kingdom DNA Methylation Dynamics: Comparative Mechanisms of 5mC/6mA Regulation and Their Implications in Epigenetic Disorders

**DOI:** 10.3390/biology14050461

**Published:** 2025-04-24

**Authors:** Yu Liu, Ying Wang, Dapeng Bao, Hongyu Chen, Ming Gong, Shujing Sun, Gen Zou

**Affiliations:** 1College of Life Sciences, Fujian Agriculture and Forestry University, Fuzhou 350002, China; 2National Engineering Research Center of Edible Fungi, Institute of Edible Fungi, Shanghai Academy of Agricultural Sciences, 1000 Jinqi Rd., Shanghai 201403, China; wyhrx@126.com (Y.W.); baodapeng@saas.sh.cn (D.B.);

**Keywords:** DNA methylation, 5mC, 6mA, epigenetic regulation, chromatin architecture

## Abstract

DNA methylation regulates gene expression and genome stability. Challenges include tissue-specific spatiotemporal heterogeneity and poorly understood non-canonical 6mA pathways. This review compares conserved vs. species-specific 5mC/6mA networks across animals, plants, and fungi by analyzing enzymes (DNMTs/TETs, CMTs/ROS1, DIM-2/DNMTAs), proposing precision epigenome editing for therapies. Critical gaps persist in fungal methylome engineering, urging mechanistic studies to harness evolutionary insights for biotechnological and biomedical advances.

## 1. Introduction

Traditional genetics holds that changes in gene expression levels stem from alterations in gene sequences, with genes determining an organism’s traits. However, the discovery of numerous biological phenomena that defy traditional genetic paradigms has demonstrated that external environmental factors can epigenetically modulate gene expression—altering phenotypic outcomes without affecting the primary DNA sequence. In 1939, British embryologist C.H. Waddington proposed the term “epigenetics” to describe heritable alterations in gene expression that occur independently of DNA sequence changes, providing a conceptual framework for these observed phenomena. Epigenetics refers to changes in gene expression and function that do not alter the DNA sequence but instead modify the structure and activity of chromatin, ultimately leading to “heritable” changes in individual phenotypes. This concept confines epigenetic regulation to the chromatin level while emphasizing its “heritable” nature [1,2]. For the genome, it is not only the sequence that contains genetic information, but its modifications can also record genetic information. DNA methylation, as a normal and widespread modification in eukaryotic cells, influences the regulation of gene expression, genetic imprinting, and transposon silencing. It is the primary epigenetic form of gene expression regulation, capable of altering genetic expression without changing the DNA sequence [3]. Extensive research has demonstrated that dysregulated DNA methylation is intricately linked to disease pathogenesis, with elucidation of its establishment mechanisms being critical for developing targeted epigenetic therapies [4,5]. CRISPR/dCas9 (dead Cas9) technology has shown great potential in multiple fields. In 2021, the p300-dCas9 system, which activates gene expression by targeting histone acetylation, was successfully constructed in *Aspergillus niger*. For the first time, CRISPR/dCas9 mediated site-specific epigenetic editing has been achieved in filamentous fungi, overcoming the limitations of traditional non-selective epigenetic interventions such as inhibitors or global knockout. This system provides programmable and high-precision tools for the study of epigenetics in filamentous fungi [6] (Figure 1).

## 2. The Role of DNA Methylation in Epigenetic Regulation: Mechanisms of Generation, Chromatin Remodeling, and Heritability

In 1975, Riggs, Holliday, and Pugh proposed DNA methylation as an epigenetic marker. DNA methylation primarily includes 4-methylcytosine (4mC), 5-methylcytosine (5mC), and N6-methyladenine (6mA), which were first discovered in the genomes of prokaryotes and single-celled eukaryotes. As research progressed, these DNA methylation modifications were also found to exist in the genomes of higher eukaryotes [7]. As the canonical DNA methylation mark in mammals, 5-methylcytosine (5mC) has been rigorously characterized for its multifaceted roles in essential biological processes—including transcriptional regulation, retrotransposon suppression, parental-origin-specific imprinting, and dosage compensation through X-inactivation [8]. This review systematically examines the enzymatic machinery governing 5mC deposition, its functional consequences on chromatin architecture, and mitotically heritable transmission mechanisms.

### 2.1. UHRF1-Mediated Catalytic Activation Mechanism of DNA Methylation

In canonical DNA methylation processes, the covalent addition of a methyl group occurs on the fifth carbon of cytosine residues when these nucleotides present as CpG dinucleotides—a designation indicating cytosine precedes guanine via a 5′-to-3′ phosphodiester bond. Regions on the DNA where CpG is densely clustered are called CpG islands. CpG islands are often found in regulatory regions such as promoters or enhancers, influencing gene transcriptional activity and genomic stability, and playing an important epigenetic role in various biological processes [9].

The methyl group for DNA methylation is generally provided by S-adenosylmethionine (SAM) and is generated and maintained under the action of methyltransferases, primarily associated with the DNMT protein family. The DNMT protein family consists of three members: DNMT1, DNMT2, and DNMT3. It was previously believed that DNMT1 could only recognize hemimethylated DNA, enabling the nascent strand to undergo methylation modification according to the methylation pattern of the template strand. The de novo methylation process was thought to be carried out by DNMT3 [10], while DNMT2 mainly acts on RNA [11,12]. In the latest research, scientists have confirmed that DNMT1 can also function as a de novo DNA methyltransferase [13]. DNMT1 is composed of multiple domains, among which the tight binding between the RFTS domain and the catalytic domain allows hmDNA to enter the catalytic site of DNMT1, resulting in the formation of an autoinhibitory structure. This structure prevents the binding of the catalytic site to hmDNA. UHRF1, as a cofactor of DNMT1, can act on the RFTS domain of DNMT1 to open up the active structure of DNMT1, playing a crucial role in the regulation of DNA methylation by DNMT1 [14] (Figure 2).

### 2.2. Synergistic Network of DNA Methylation and ATP-Dependent Remodeling Complexes

Chromosomes enable chromatin to dynamically transition between a condensed state and a transcriptionally accessible state. By adjusting the chromatin structure, the accessibility of certain gene regions is increased or decreased, thereby influencing the binding of transcription factors and RNA polymerase, ultimately regulating gene expression [15]. This transition is achieved through covalent modifications of chromatin components (such as histone acetylation, and DNA methylation) and non-covalent modifications (such as the action of ATP-dependent chromatin remodeling factors). Non-covalent modifications primarily rely on four major protein families: the SWI/SNF family, also known as the BAF complex, is believed to alter nucleosome positioning and structure, thereby regulating gene expression. The CHD ATPase family has a labeled chromatin domain that binds to methylated lysine residues [16]. The ISWI family regulates nucleosome sliding and spacing [17]. INO80 ejects nucleosomes to assist in the repair of double-strand breaks, allowing repair factors to access the DNA [18]. These four families work together to influence the gene transcription process. DNA methylation can cause changes in the chromatin structure of corresponding regions in the genome, leading to highly spiralized chromatin that condenses into clumps and loses transcriptional activity. At the same time, an increase in DNA methylation levels also affects the non-covalent modifications of chromatin remodeling, indicating that these processes are not independent. Knocking out genes related to demethylation results in an overall increase in methylation levels, and the expression of both the SWI/SNF family complex and the INO80 family complex is affected [19].

### 2.3. Epigenetic Stability and Heritability

During cell division and replication, genomic information is passed on to daughter cells, and epigenetic modification information is also transmitted to maintain the cell’s state. Unlike the genome, the transmission of epigenetic information across generations is imprecise yet stable, and highly plastic, capable of adapting to and maintaining different initial epigenetic states [2]. Research has found that the imprecise methylation maintenance activity and low-level de novo methylation activity of DNMT1 lead to random DNA methylation. These epigenetic mutations tend to be repaired by a correction mechanism sensitive to the methylation levels of adjacent regions during cell expansion, possibly through the imprecise activity of DNMT1. Moreover, this imprecise activity can also achieve a stable inheritance of different initial methylation levels, thereby simultaneously realizing the plasticity and heritability of epigenetic modifications. This allows the epigenome to change with the transformation of cell fate and ultimately helps maintain the specific state of the cell [20].

## 3. Multi-Mechanism Synergistic Regulation of DNA 5mC Demethylation

During transcription, transcription factors (TFs) regulate the expression of target genes by binding to specific sites within their enhancers and promoters. The binding sites of most TFs contain CpG dinucleotides. When multiple methylation modifications occur on the promoter, it becomes difficult for the transcription factor to recognize the promoter it seeks, leading to the silencing of gene expression. The removal of methyl groups from these promoters is essential for TF binding and gene expression [21,22,23]. In mammals, DNA 5mC demethylation primarily relies on the TET-TDG-BER active demethylation system, and subsequent research has also uncovered other demethylation mechanisms.

### 3.1. TET-TDG-BER-Mediated DNA Demethylation Mechanism

Methylation of cytosine within CpG dinucleotides is a common epigenetic modification of DNA, dynamically established and maintained by DNA methyltransferases and demethylases. The molecular mechanism of active DNA demethylation has only recently begun to emerge with the discovery of 5mC directed hydroxylases, TET proteins, and the base excision activity of Thymine DNA Glycosylase (TDG) [24].

In mammals, DNA methylation predominantly occurs in symmetrical CG cytosine contexts, accounting for about 70–80% of all CG sites across the human genome [25]. Methylcytosine, recognized as the fifth base encoding epigenetic information, also poses a significant risk as an endogenous mutation source due to its inherent instability. Methylcytosine undergoes spontaneous deamination at a rate significantly higher than that of cytosine, leading to the production of thymine [26]. The process of active DNA demethylation primarily relies on the stepwise oxidation by the Ten-Eleven Translocation (TET) family of dioxygenases. The TET family consists of three members: TET1, TET2, and TET3. All three TET enzymes exhibit Fe^2+^- and α-ketoglutarate (α-KG)-dependent dioxygenase activity, initiating DNA demethylation by converting 5mC to 5-hydroxymethylcytosine (5hmC), and further oxidizing 5hmC to 5-formylcytosine (5fC) and 5-carboxylcytosine (5caC) [27]. The dioxygenase domains of TET1, TET2, and TET3 are composed of a cysteine-rich (Cys) domain and a tightly packed double-stranded β-helix (DSβH) fold domain, which mediate catalytic activity. The DSβH domain includes three Fe^2+^ binding sites and one α-KG binding site. Additionally, full-length TET1 and TET3 proteins contain an N-terminal CXXC-type zinc finger [28,29]. The CXXC-type zinc finger domain stabilizes a short, self-folding polypeptide spatial configuration that can adopt a “finger” shape through the binding of Zn^2+^ to characteristic groups of amino acid residues in the peptide chain, enabling interaction with certain DNA/RNA. The CXXC domain regulates the binding of TET1 and TET3 to DNA sequences, whereas the TET2 protein lacks a DNA recognition domain and requires interaction with other DNA-binding proteins to engage with DNA [30].

After the stepwise oxidation of methylated cytosine, DNA glycosylases remove damaged or modified nucleobases by cleaving the N-glycosidic bond and restoring the correct nucleotide through subsequent base excision repair. In addition, DNA glycosylases also contribute to epigenetic regulation by mediating DNA demethylation and performing other crucial functions [31]. The clearance of methylation relies on TDG, a monofunctional glycosylase that initiates BER by recognizing specific sequences, including 5mC deaminated to form G:T mismatches or oxidized by the TET enzyme family to 5fC and 5caC [32,33,34]. Protein interaction results indicate that TET1 and TDG physically interact through specific N-terminal and C-terminal TET1 domains [24]. TDG catalyzes the excision of 5fC and 5caC, generating an apurinic/apyrimidinic (AP) site. By coordinating with BER enzymes, TDG mediates the replacement of 5fC and 5caC with cytosine [30].

TDG exhibits robust in vitro base excision activity towards 5fC and 5caC paired with G in double-stranded DNA [35]. Using immunoprecipitation in TDG-deficient ES cells, significant enrichment of 5fC and 5caC in non-repetitive regions, particularly at distal regulatory elements, can be observed [36,37]. Additionally, simultaneous overexpression of TET and TDG in HEK293 cell lines leads to the loss of TET-associated 5caC or 5fC [38]. The excision function of TDG requires the instability of the N-glycosidic bond, a condition that 5fC and 5caC meet more readily than C, 5mC, and 5hmC [39,40]. TDG possesses structural features that mediate the recognition of these oxidized C bases, including a binding pocket specifically designed to accommodate the 5-carboxyl substituent [41]. The process of active demethylation results in an overall loss of 5mC [42,43,44,45]. Following the initial passive dilution of 5mC, 5hmC subsequently accumulates actively and then is lost in a replication-dependent manner. In fact, TDG has a higher affinity for 5fC and may employ different mechanisms in the excision of 5fC and 5caC [32,46].

Base Excision Repair (BER) is the final step in active demethylation. BER plays a crucial role in the genome-wide active DNA demethylation of Primordial Germ Cells (PGCs) [47]. 5fC is recognized and removed by TDG to create an abasic site. In humans, BER is initiated by one of the 11 damage-specific DNA glycosylases, which recognize the damage and cleave the glycosidic bond to leave an abasic site. Repair is completed through the action of several proteins that restore the correct base pair [48]. Within cells, ATP-ribosyltransferases PARP1 and PARP2 contribute to DNA Base Excision Repair and DNA demethylation. BER proteins are covalently poly-ADP-ribosylated by PARP, but the role of this post-translational modification in the BER process is not yet clear. PARP senses the AP sites and Single-Strand Breaks (SSBs) generated during TET-TDG mediated active DNA demethylation and covalently attaches PAR (Protease-Activated Receptor) to each participating BER protein. Covalent PARylation dissociates BER proteins from the DNA, thereby accelerating the completion of the repair process [49].

### 3.2. Dual Roles of the DNMT Protein Family in DNA Methylation and Demethylation

The DNMT protein family influences the process of methylation. Passive demethylation primarily involves inhibiting the cell’s methylation maintenance mechanisms, largely dependent on the DPPA3 protein [50]. DPPA3 can bind to UHRF1 [51], a crucial protein involved in the methylation maintenance mechanism. By preventing UHRF1 from participating in DNA methylation maintenance, DNA can be passively demethylated. Under conditions of extremely low SAM levels, DNMT3A and DNMT3B may also demethylate 5mC by converting it to thymine through deamination, which is subsequently replaced by unmodified cytosine via the base excision repair pathway [52]. Alternatively, when 5mC is converted to 5hmC by TET enzymes, it can be further modified to unmodified cytosine under oxidative redox conditions by DNMT-3a and DNMT-3b [53]. TDG has been shown to interact with numerous transcription factors, chromatin-modifying enzymes, and DNMTs, increasing the likelihood that TDG plays a functional role in regulating gene transcription through its glycosylase activity or as a transcriptional coactivator [54]. Both the PWWP domain and the catalytic domain of DNMT3a can mediate its interaction with TDG at the N-terminus. This interaction affects the enzymatic activities of both proteins: DNMT3a positively regulates TDG’s glycosylase activity, while TDG inhibits DNMT3a’s methylation activity in vitro [26]. Although DNMTs play a role in active demethylation, the exact pathways have not yet been fully elucidated. The mechanism of DNA methyltransferases may differ from that of active demethylation. Studies have reported that nitric oxide can inhibit the DNA demethylases TET and ALKBH2, but it has no effect on the activity of methyltransferases [55].

### 3.3. AID/APOBEC-Mediated Oxidative Deamination-Demethylation Pathway

The activation-induced cytidine deaminase (AID)/APOBEC family mediates another oxidative deamination-demethylation pathway. AID initiates antibody diversification in germinal center B cells by deaminating cytosine, leading to somatic hypermutation and class switch recombination. AID-mediated deamination is believed to cause active demethylation of 5-methylcytosine in DNA [56,57]. AID primarily catalyzes the deamination of cytosine at 5hmC sites, generating 5-hydroxymethyluracil (5hmU). 5hmU is subsequently excised by TDG, single-strand-selective monofunctional uracil DNA glycosylase 1 (SMUG1) recombinant protein, NEIL1 antigen (recombinant protein), or methyl-CpG-binding protein 4 (MBD4), and replaced by cytosine through the action of BER enzymes [58,59].

### 3.4. GADD45a/b-Mediated Active DNA Demethylation Mechanism

Growth arrest and DNA damage-inducible proteins 45a (GADD45a) and 45b (GADD45b) also play crucial roles in the demethylation of specific promoters [60,61,62]. R-loops, which are DNA-RNA hybrids enriched at CGIs, can modulate chromatin states [63,64]. GADD45a directly binds to R-loops and mediates local DNA demethylation by recruiting TET1. Oxidized DNA demethylation intermediates are enriched at genomic R-loops, and their levels increase with the depletion of RNase H1. Studies suggest that GADD45a acts as an epigenetic R-loop reader, facilitating CGI demethylation. Further research indicates that TDG, AID, and GADD45a form a ternary complex that regulates the methylation status of genomic promoters and enhancers. Therefore, it is likely that GADD45a/b-TDG-AID-BER collectively mediates active DNA demethylation [65].

### 3.5. Direct Decarboxylation Pathway of 5caC and the Alternative Role of UNG2 in Mammalian Cell Demethylation

Researchers designed and synthesized DNA containing 5caC and, by combining stable isotope labeling with mass spectrometry analysis, discovered that 5caC in DNA can be directly decarboxylated to form cytosine through the cleavage of a carbon-carbon bond. This constitutes a novel pathway for the demethylation of 5mC in DNA. This pathway involves only a single decarboxylation step following the oxidation of 5mC to 5caC by TET proteins, providing significant insights into the regulatory mechanisms of DNA epigenetic modifications [49]. TDG is essential for active demethylation in embryonic stem cells and induced pluripotent stem cells, but it is rarely expressed in mouse zygotes. Using a methylated luciferase reporter gene assay for functional genomic screening, researchers identified other factors that may contribute to demethylation in mammalian cells. UNG2, one of the known glycosylases that remove uracil residues from DNA, was found to reduce DNA methylation and activate transcription of methylation-silenced reporter genes when co-transfected with TET2 into HEK293T cells. Additionally, UNG2 can reduce genomic 5caC levels in transfected cells, similar to TDG [66].

### 3.6. Dual Functions of the Methyl-CpG Binding Domain IV (MBD4)

Methyl-CpG binding domain IV (MBD4) also functions as a methyl repair enzyme to counteract the mutagenic effects of methylated cytosine. The protein encoded by this gene is a member of the nuclear protein family associated with the presence of the methyl-CpG binding domain (MBD). These proteins are capable of specifically binding to methylated DNA, and some members can also suppress the transcription of methylated gene promoters. The protein contains an MBD domain at the N-terminus, which plays a role in binding to methylated DNA and protein interactions, as well as a C-terminal mismatch-specific glycosylase domain involved in DNA repair [31].

### 3.7. DNA Methylation Reprogramming During Early Embryonic Development

Both sperm and egg cells are highly differentiated cells with high levels of DNA methylation, while the fertilized egg has a very low level of methylation. After fertilization, both sperm and the fertilized egg undergo extensive DNA demethylation, establishing the totipotency of the early embryo. Taking mice as an example, the DNA dioxygenase TET3 mediates active demethylation, while the glycosylase TDG is not involved in this process [67]. As research on DNA methylation reprogramming deepens, scientists have discovered that human primordial germ cells share key characteristics with mouse primordial germ cells but also exhibit unique differences. Human primordial germ cells also undergo extensive DNA methylation erasure during development, with DNA methylation levels reaching their lowest point around weeks 10–11 of embryonic development. Although DNA methylation is completely erased in most regions of the human primordial germ cell genome, significant DNA methylation remains on some repetitive sequence elements. Such a large-scale methylation reprogramming process likely involves the crucial roles of other key components of epigenetic regulation, particularly various covalent modifications of histones [68]. Subsequent research has revealed that from the union of sperm and egg to the blastocyst stage, DNA 5mC methylation levels show a downward trend. Considering that active demethylation is the primary pathway for demethylation, passive demethylation is not considered the main mechanism in DNA reprogramming. Based on the known process of active DNA demethylation, 5mC is oxidized to 5hmC for subsequent processes, and during this stage, 5hmC levels should theoretically increase. However, recent studies have found that from the fertilized egg to the blastocyst, 5hmC levels consistently decrease [69]. This suggests that 5hmC may have other biological functions during DNA methylation reprogramming [70].

## 4. Epigenetic Regulatory Network of 6mA Demethylation

6mA modification is a prevalent epigenetic marker in prokaryotes. However, due to the low abundance of 6mA modification in eukaryotes, studying its effects has been challenging. In recent years, technological advancements have led to the discovery of 6mA in higher eukaryotic genomes [71]. Unlike 5mC, which increases DNA helix stability and suppresses gene expression, 6mA disrupts DNA helix stability, leading to DNA unwinding. 6mA primarily occurs at ApT dinucleotides near transcription start sites (TSS) and is associated with active gene expression [72]. In the human genome, 6mA constitutes approximately 0.051% and is enriched in coding regions, marking actively transcribed genes in human cells [73]. Abnormal changes in 6mA levels have been observed in certain diseases. Based on the above findings, we will discuss 6mA modification and its impact on diseases.

### Dynamic Modification and Demethylation Mechanisms of 6mA

Similar to 5mC methylation, DNA 6mA methylation primarily involves the addition of a methyl group to DNA using SAM as the donor, facilitated by specific DNA adenine methyltransferases, mainly the MT-A70 protein family [74]. The detailed mechanisms of 6mA methylation remain unclear.

The removal of 6mA modifications can occur through two main pathways: first, via Fe(II) and α-ketoglutarate-dependent dioxygenases of the ALKB family; second, through the conversion of 6mA to hypoxanthine by 6mA deaminase, followed by base excision repair mediated by hypoxanthine DNA glycosylases of the ALKA family [7]. Although the presence of 6mA in mammalian genomic DNA (gDNA) is debated [75], ALKBH1 has been identified as a 6mA demethylase in human mitochondria [76,77]. ALKBH1 mediates 6mA demethylation in single-stranded DNA (ssDNA) or bulged DNA, with the final repair completed by the base excision repair mechanism. This process is analogous to the active demethylation of 5mC via the TET-TDG-BER system. 6mA is oxidized to 6hmA with the help of α-ketoglutarate and Fe^2+^, and the aldehyde group is then excised and repaired back to 6mA. In human mitochondria, 6mA regulates mitochondrial function, and the loss of ALKBH1 reduces mitochondrial oxidative phosphorylation [78]. In *Drosophila melanogaster*, DMAD has been identified as a 6mA demethylase. DMAD removes 6mA from transposon regions, which is associated with transposon suppression. Based on its sequence, DMAD contains conserved Fe^2+^ and 2-oxoglutarate-dependent dioxygenase domains and a DSBH domain, showing homology to bacterial ALKB proteins and mediating 6mA demethylation [79]. However, in another study on *D. melanogaster*, DMAD was shown to maintain active transcription by regulating intragenic 6mA in genes involved in neural development and neuronal function. These contrasting results suggest that 6mA located in different regions and gene clusters may lead to different transcriptional outcomes [80]. In *Caenorhabditis elegans*, the enzyme responsible for 6mA removal is NMAD-1, which is homologous to members of the ALKB family [81]. Additionally, ALKBH4 in *C. elegans*, homologous to the *D. melanogaster* 6mA demethylase, also exhibits 6mA demethylation activity [82].

In prokaryotes, the removal of 6mA can be divided into two steps: deamination and excision. First, 6mA is hydrolyzed by deaminase to generate hypoxanthine, which is recognized as a damaged base by the ALKA enzyme, leading to the cleavage of the glycosidic bond. Subsequently, with the assistance of AP endonuclease, the residual 5′ deoxyribose phosphate group is exposed, and finally removed by deoxyribose phosphate diesterase. The DNA repair is then completed through the action of DNA polymerase I and DNA ligase [83] (Figure 3).

## 5. Epigenetic Regulation of DNA Methylation and Demethylation in Plants

In plants, cytosine methylation can occur in all sequence contexts, including symmetric CG and CHG contexts, as well as asymmetric CHH contexts (where H = A, T) [25]. MET1 is the first discovered plant methyltransferase, maintaining CG methylation in gene-coding regions. CMT is a plant-specific methyltransferase [84,85]. The mechanisms of active DNA demethylation in plants differ significantly from those in animals [86]. Below, we discuss active demethylation in plants and compare it with active demethylation in animals.

### 5.1. Molecular Mechanisms of Active DNA Demethylation in Plants

Unlike in mammals, active demethylation in plants primarily occurs through DNA glycosylases, which hydrolyze the glycosidic bond between the base and its deoxyribose residue and cleave the DNA backbone at the abasic site, removing methylated cytosines and creating single-nucleotide gaps. DME and ROS1 are bifunctional DNA glycosylases involved in the base excision repair (BER) pathway [87]. DME and ROS1 catalyze β-elimination or successive β, δ-elimination reactions when cleaving the DNA backbone. The β-elimination reaction produces 3′-PUA, while the successive β, δ-elimination reaction produces 3′-phosphate. Both 3′-phosphate and 3′-PUA must be converted to 3′-hydroxyl (3′-OH) by zinc finger DNA 3′-phosphatase (ZDP) to allow DNA polymerase and ligase activities to fill the gap. It is worth noting that although active DNA polymerase in plants has not yet been confirmed, the DNA ligase responsible for sealing the gap has been identified as *AtLIG1* [88]. APE1L (an AP endonuclease, homologous to human APE1) functions in *Arabidopsis*, efficiently processing 3′-PUA to generate 3′-OH, similar to the role of ZDP [89,90,91]. Dysfunction of ZDP leads to DNA hypermethylation at approximately 1500 endogenous sites [92] (Table 1).

DML2 has been identified as a DNA demethylase in tomatoes, and the absence of DML2 leads to an increase in DNA methylation levels throughout the tomato genome, RIN and FUL1 are key transcription factors in the tomato ripening process. DNA methylation affects tomato fruit ripening by inhibiting the expression of RIN and FUL1 [93]. In *Arabidopsis*, there are four DNA demethylase-encoding genes, including *AtROS1*, *AtDME*, *AtDML2*, and *AtDML3* [87]. Dysfunction of ROS1 leads to DNA hypermethylation at the RD29A promoter, resulting in the silence of RD29A-luc [94]. In mutants of ROS1, DML2, and DML3, DNA methylation increases in both *Arabidopsis* and tomatoes [95,96,97]. Additionally, DME and ROS1 can also remove thymine from T:G mismatches [91]. The protein complexes IDM and SWR1 regulate ROS1 function. The DNA methylation-binding protein MBD7 within the IDM complex binds to highly methylated genomic regions and recruits other anti-silencing factors, IDM1, IDM2, and IDM3, to form a protein complex. IDM1 catalyzes histone acetylation, creating a chromatin environment favorable for *AtROS1* activity. MBD9 and NPX1 in the SWR1 complex can bind to acetylated histones catalyzed by IDM1, thereby promoting the SWR1 complex to regulate the accumulation of H2A.Z, which can interact with ROS1 to regulate DNA demethylation [98] (Figure 4).

### 5.2. Distribution Analysis of 6mA Demethylases in Plants

6mA has also been detected in the *Arabidopsis* genome [99]. 6mA sites are widely distributed and enriched in pericentromeric heterochromatic regions, showing a positive correlation with gene expression in *Arabidopsis*. In rice, *OsALKBH1* has been identified as a 6mA demethylase based on its sequence homology to mouse ALKBH1, and it has been found that disruption of *OsALKBH1* increases 6mA levels [100]. Structural analysis of *OsALKBH1* reveals the presence of a DSHB domain, and mutations at its enzymatic site result in the loss of demethylase activity [101].

## 6. Epigenetic Regulatory Network of DNA Methylation in Fungi

DNA methylation in fungi occurs in both coding and non-coding regions, with lower abundance compared to mammals and plants. Initially, it was thought to be specific to transposable elements (TEs), where it exerts its suppressive function through interactions between transcription factors and DNA. As the methylation level of TEs increases, their expression capacity decreases [102].

Repetitive sequences are prevalent in eukaryotic genomes [103]. In fungi, methylation is notably biased toward CG sites in repetitive sequences, while lower methylation levels are found in gene-coding regions [104]. The loss of the methyltransferase gene Dim2 in the wheat leaf blight pathogen reduces the mutation rate of repetitive sequences, further demonstrating the impact of 5mC methylation on repetitive sequences [105]. This phenomenon was first observed in *Coprinopsis cinerea*, where changes in methylation of repetitive sequences occur during the dikaryotic phase of sexual reproduction [106]. In the filamentous fungus *N. crassa*, DNA methylation mainly occurs in the constitutive heterochromatin region, which contains highly repetitive DNA sequences and usually lacks transcriptional activity, making it an ideal model for studying DNA methylation [107].

### 6.1. From Epigenetic Regulation by the DNMT/Rad8 Family to Molecular Mechanisms of Pathogenic Fungal Virulence

Most fungi possess at least one methyltransferase (from the DNMTs family or Rad8 family). In basidiomycetes, each genome contains only one copy of the DNMT2 and Rad8 subfamily genes, but multiple copies of the DNMT1 subfamily. Homologs of DNMT1 and DNMT2 are present in both ascomycetes and basidiomycetes, while Rad8 proteins are found in basidiomycetes and ascomycetes but not in zygomycetes [108]. DNMTs in *Trichoderma reesei* affect genome-wide cytosine methylation and directly participate in meiotic DSB initiation and repair. RID1 regulates the initiation of type I DSB, which is necessary for Rad51 mediated DSB repair and normal meiosis. DIM2 and RID1 have partially redundant functions but do not directly participate in the repair of Rad51 dependencies [109].

Fungal DNA methylation is associated with various lifestyles, such as decomposition, parasitism, and life cycles, making the study of fungal DNA methylation more complex [110]. In most fungi, DNA methylation is lower in the mycelial stage than in the conidial stage. However, in *Beauveria bassiana* and *Metarhizium robertsii*, the opposite is true. In *B. bassiana*, the overall methylation levels in genes and transposable element (TE) regions are higher in the mycelial stage than in the conidial stage [111]. In *M. robertsii*, approximately 0.38% of cytosines are methylated in conidia, compared to about 0.42% in mycelia, suggesting a methylation reprogramming pattern similar to that in mammals. Studies have proposed *MrDIM-2* and *MrRID* as 5mC methyltransferases in *M. robertsii*, although the specific molecular mechanisms remain unclear [112]. *Cordyceps militaris*, an entomopathogenic fungus with significant medicinal and edible value, often exhibits degeneration during subculturing and preservation. Studies have found that DNA 5mC levels are higher in degenerated strains, while the methyltransferase *CmMAT* is lower than in normal strains [113]. In the *Ganoderma lucidum* genome, approximately 1.8% of cytosines are methylated [114]. In *Laccaria bicolor* and *C. cinerea*, 5mC levels are similar, concentrated in exons and intergenic regions, at around 0.9%. The levels of 5hmC, 5fC, and 5caC decrease sequentially, with enrichment in TEs and intergenic regions. However, L. bicolor has 4–5 times higher levels of 5hmC and 5fC than *C. cinerea*. In *C. cinerea*, 5hmC shows stronger enrichment on TET genes [115], while 5caC levels are extremely low in both fungi, at about 0.0001% [116]. In *C. cinerea*, a mammalian TET homolog (CC1G_05589, *CcTET*) has been reported, displaying activity similar to its mammalian counterparts. Its biological function is also analogous, capable of progressively oxidizing 5mC to 5hmC, 5fC, and 5caC in *C. cinerea* [117]. Its molecular mechanism is identical to that in mammals, providing a valuable model for studying active DNA demethylation in other fungi. Heterochromatin silencing is regulated at multiple levels in *N. crassa*. Chromatin remodeling factor (DIM-1) plays a crucial role in maintaining heterochromatin structure and epigenetic regulation. DIM-1 mutations lead to DNA methylation defects, as well as abnormal nucleosome spacing across the entire genome, accompanied by gene expression disorders. The normal arrangement of nucleosomes is a prerequisite for the functioning of heterochromatin mechanisms such as H3K9me3 and DNA methylation [118]. DNA methyltransferase deficient protein 2 (DIM-2) relies on heterochromatin markers H3K9me3 and HP1 to exert methylation activity, and the two synergistically guide the targeted methylation of DIM-2 in heterochromatin [119]. DIM-5 (H3K9 methyltransferase) is the core catalytic subunit for DNA methylation, and its localization and activity depend on DIM-7. Although DIM-10 (a homolog of *Saccharomyces cerevisiae* Clr5) is not essential for DNA methylation, it synergistically inhibits heterochromatin region genes through methylation [107]. The DNA methylation regulation of *T. reesei* for industrial use is complex. In production, epigenetic abnormalities often lead to spontaneous loss of cellulase productivity, resulting in economic losses. DNA methylation inhibits cellulase production, and reducing methylation levels can partially restore enzyme production. Knocking out DNA methyltransferase (Dim2/Rid1) can restore cellulase production in the short term, but it still loses cellulase activity after long-term passage [120].

In the plant pathogenic fungus *Verticillium dahliae*, the establishment of DNA methylation patterns mediated by DNA methyltransferases positively regulates fungal virulence. The epigenetic mechanism of DNA methylation modulates the virulence of plant pathogenic fungi [121]. In *Cryphonectria parasitica*, *CpDmt1* and *CpDmt2* are proposed as methyltransferase genes influencing mycelial growth [122]. HOA (homoserine O-acetyltransferase) is a potential antifungal target in *Magnaporthe oryzae*. Deletion of the HOA-encoding genes *MoMET2* and *MoCYS2* disrupts the biosynthesis of the methyl donor S-adenosylmethionine and severely impairs the development and virulence of *Aspergillus oryzae*. In ∆*Momet2* mutants, 5mC modifications significantly increase, suppressing the expression of genes required for pathogenicity [123]. *Botrytis cinerea* is one of the most destructive fungal pathogens. The DNA methyltransferases *BcDIM2* and *BcRID2* exhibit strong synergistic effects in regulating the pathogenicity of *B. cinerea*. When both are knocked out, only a few 5mC sites are detected in *B. cinerea* and pathogenicity toward various hosts is completely lost [124].

Epigenetic editing based on CRISPR/Cas9 in fungi has become an important research direction at the intersection of synthetic biology and epigenetics in recent years. The CRISPR/Cas9 system targets specific genomic loci through gRNA and utilizes inactivated Cas9 (dCas9) to fuse with epigenetic modifying enzymes (such as DNA methyltransferase and histone modifying enzymes) to regulate gene expression without altering the DNA sequence [125]. In the *A. niger* study, target genes (*flbA* and *GFP*) were successfully silenced in *A. niger* by fusing dCas9 with ZNF10 KRAB and DNMT3A/DNMT3L domains. DNA methylation was stably expressed in *A. niger* gene silencing and exhibited high stability in genetic processes. The dCas9-TET1 fusion protein (CRISPRon) can actively remove DNA methylation and fully restore *flbA* expression, with a phenotype consistent with the wild type. This study is the first to transplant CRISPRoff/CRISPRon from human cells to fungi, verifying its cross-species applicability [126]. In 2024, Takehiko Todokoro et al. improved the targeted knock-in efficiency of *A. oryzae* using CRISPR-Cas9 technology, making it easier to achieve gene editing in *A. oryzae* [127]. In 2025, researchers developed a continuous evolution platform for filamentous fungi using a cytidine base editor (CBE) based on CRISPR-Cas9, which solved the problem of traditional tools having difficulty achieving complex phenotype iterative optimization, marking a breakthrough application of synthetic biology tools in the rational evolution of industrial strains [128].

### 6.2. Epigenetic Regulation of DNA N6-Methyladenine (6mA) in Fungi

6mA holds significant biological and evolutionary importance in early-diverging fungi (EDF), which exhibit diverse evolutionary patterns in 6mA utilization [129]. Species with high 6mA levels display symmetric methylation enriched in highly expressed genes, while those with low 6mA levels predominantly show asymmetric 6mA. A specific methyltransferase responsible for asymmetric 6mA methylation has been identified in EDF, whereas the MTA-70 complex is responsible for symmetric methylation [130]. In fungi, 6mA occurs symmetrically at ApT dinucleotides and is concentrated in dense methylated adenine clusters around the transcription start sites of expressed genes. Its distribution is negatively correlated with that of 5mC [112]. In the nuclear DNA of basal fungi, 6mA modifications are relatively abundant, primarily located near transcription start sites and positively correlated with transcription [131]. In algae and mucoromycetes, 6mA levels are approximately 1.13% and 0.26%, respectively [130]. *B. cinerea*, an important pathogenic fungus, exhibits 6mA primarily concentrated in upstream promoter regions. BcMETTL4 is the 6mA methyltransferase in *B. cinerea*. When BcMETTL4 activity is reduced, 6mA levels significantly decrease, and the virulence of *B. cinerea* is consequently diminished [132]. In mucoromycetes, Dmt1, Dmt2, and Dmt3 encode proteins similar to *C. elegans* NMAD-1 and ALKBH1 [133]. CcTET, initially identified as a 5mC demethylase in *C. cinerea*, has also been recognized as a 6mA demethylase involved in 6mA demethylation in *C. cinerea* [134]. Its mechanism resembles the active 6mA demethylation mechanism in mammals. Under the action of CcTET, 6mA modifications in *C. cinerea* are oxidized to 6hmA, which is ultimately converted to normal adenosine. Structural analysis of CcTET reveals that it utilizes three flexible loop regions and two key residues (D337 and G331) in its active pocket to preferentially recognize substrates on dsDNA. Interestingly, the CcTET D337F mutant retains catalytic activity for 6mA but loses activity for 5mC [68,134]. This indicates that although both 5mC and 6mA undergo oxidation of methylated bases under the action of dioxygenases during active demethylation, their mechanisms are distinct [135]. BcMETTL4, the 6mA methyltransferase in *B. cinerea*, undergoes point mutations that lead to a significant reduction in 6mA levels and a decrease in the fungus’s virulence. 6mA provides a potential epigenetic marker in *B. cinerea*, and BcMETTL4 regulates the virulence of this important plant pathogen [128]. In *B. cinerea*, the absence of both 5mC and 6mA methyltransferases significantly impacts its virulence, highlighting the crucial role of methylation modifications in the pathogenicity of fungal pathogens. 6mA regulates essential processes in mucoromycetes, including DNA replication or repair, virulence, and lipid biosynthesis. Additionally, studies have found that light response also modulates 5mC and 6mA levels in mucoromycete mycelia, showing an inverse relationship between the two modifications [132]. 6mA regulates critical processes in *Mucor*, including DNA replication or repair, virulence, and lipid biosynthesis. Additionally, studies have found that light response also modulates the levels of 5mC and 6mA in Mucor mycelia, resulting in an inverse correlation between the two modifications [130].

## 7. Conclusions

Both DNA 5mC and 6mA modifications utilize SAM as the methyl donor during their formation. Despite being forms of methylation, the proportion of 5mC in most organisms is significantly higher than that of 6mA [106]. Two hypotheses are proposed to explain this phenomenon: First, from a kinetic perspective, the energy required for 5mC formation might be lower, making 5mC more readily formed. It has been reported that CpG sequences appear to be the preferred targets for methylation in *C. cinerea* [136]. Second, there may be a competitive relationship between the two modifications, where an increase in one leads to a decrease in the other. Studies have already demonstrated a negative correlation between the distribution of 5mC and 6mA in fungi [112].

In current research, the active demethylation process of 5mC in organisms is more thoroughly understood. Upon closer examination, the active demethylation process of 6mA shares many similarities. In *C. cinerea*, the 5mC demethylase CcTET has also been shown to possess the ability to actively demethylate 6mA [134]. In mammalian mitochondria, both the 6mA demethylase ALKBH1 and the 5mC demethylase TETs oxidize methyl groups to hydroxymethyl groups with the assistance of α-ketoglutarate and Fe^2+^ [137]. Both 5hmC and 6hmA have been confirmed to be repaired to normal bases through the excision of aldehyde groups. However, unlike 6hmA, 5hmC can be further oxidized and excised by TETs. Although it has been mentioned that the demethylation mechanisms of 5mC and 6mA in fungi are not identical [135], it remains to be discovered whether 6hmA undergoes further oxidation to form analogs of 5fC and 5caC.

Methyltransferases are essential for DNA methylation, and higher methyltransferase activity facilitates DNA methylation. In fungi, for example, in *C. militaris*, degenerated strains exhibit higher DNA 5mC levels (0.56%) compared to normal strains (0.48%) [113], despite having lower methyltransferase activity. This leads to the hypothesis that as SAM, the methyl donor, is continuously consumed, once methylation levels reach a high threshold, methyltransferases may no longer need to remain highly active to maintain these levels [138]. Thus, in degenerated *C. militaris* strains, high methylation levels coexist with reduced methyltransferase activity.

Some edible and medicinal fungi, such as *C. militaris* and *G. lucidum*, are beneficial to human health and are highly valued in consumer markets. However, the degeneration of strains during subculturing has long been a concern. In *C. militaris*, degenerated strains exhibit 16.7% higher methylation levels than normal strains, prompting us to consider the link between fungal degeneration and excessive methylation levels [113]. In agricultural production, incorporating methylation level detection as a screening criterion and selecting low-methylation varieties for cultivation could enhance economic efficiency.

Currently, several enzymes with methylation and demethylation capabilities have been identified in fungi, but their specific molecular mechanisms remain unclear and require further reference to mammalian methylation patterns. Research progress in fungi is still incomplete, partly due to the low abundance of methylation in fungi [139]. Currently, the sequences of 2000–3000 fungal genes have been published. The genomes of certain parasitic or symbiotic fungi are relatively small, approximately 2.5–3 Mb, while the genome of yeast is around 12–15 Mb. Most filamentous fungi have genomes ranging from 25–50 Mb, and some basidiomycetes can have genomes as large as 50–150 Mb or even larger, containing a significant number of repetitive sequences and transposable elements [140]. The number of genes generally ranges between 5000 and 15,000. However, the gene count varies among different fungal species. For example, both *S. cerevisiae* (baker’s yeast) and *N. crassa* have fewer than 15,000 genes. In comparison, the human genome is approximately 3200 Mb in size, with 25,000 to 30,000 genes [141]. The vast differences in genome sizes suggest that fungal chromosomes are more dynamic. Additionally, the diverse lifestyles of fungi, including decomposition, parasitism, and complex life cycles, further complicate research efforts [134]. In recent years, CRISPR/Cas9 technology has been widely used in the field of fungal gene editing due to its high efficiency and precision. Traditional CRISPR/Cas9 achieves gene knockout or insertion by cutting DNA double strands, while epigenetic editing achieves gene silencing or activation by fusing dCas9 with epigenetic modifying enzymes (such as DNA methyltransferase, histone acetyltransferase/deacetylase), without altering the DNA sequence, to regulate the chromatin state of target genes [140]. This technology provides new strategies for fungal functional genomics research, regulation of secondary metabolites, and prevention and control of pathogenic fungi. In industrial fungi, targeted epigenetic modifications can dynamically regulate the synthesis gene clusters of antibiotics or enzymes, significantly increasing product yield; In plant pathogenic fungi, epigenetic silencing of virulence-related genes can reduce their pathogenicity [120]. Compared to traditional gene editing, epigenetic editing has the advantages of reversibility, efficiency, and avoiding the accumulation of gene mutations, especially suitable for complex phenotype regulation and research on host-pathogen interactions. The improvement of fungal CRISPR epigenetic editing technology will provide great assistance for fungal research and industrial applications [142].

DNA methylation is essential for the normal functioning of organisms, and its necessity is undeniable. The stability of DNA methylation inheritance aids in the transmission of genetic information while also providing epigenetic modifications with significant plasticity [143]. Under the influence of methyltransferases and demethylases, methylation levels continuously fluctuate, inevitably leading to negative effects. The occurrence of certain diseases is often accompanied by abnormal methylation at specific genomic loci, reflecting the plasticity of methylation. However, the stable inheritance of these modifications poses significant challenges to organismal health. It is often said that neither too much nor too little DNA methylation is beneficial [144]. Finding the optimal level of methylation and achieving controllable genomic methylation levels remain areas requiring extensive research. Currently, it appears that this is not determined by a single gene but involves considerations of histone methylation, histone acetylation, and other factors [145,146]. Current research on 5mC/6mA methylation indicates that there are both similarities and differences among different species. Although both DNMTs in mammals and CMTs in plants can maintain genomic stability, there are differences in their domain composition [56,85]. DIM-2 in fungi and DNMT3 in mammals have similar functions, suggesting that different organisms may have developed similar strategies to cope with genetic regulation over the long course of evolution [26,112].

## Figures and Tables

**Figure 1 biology-14-00461-f001:**
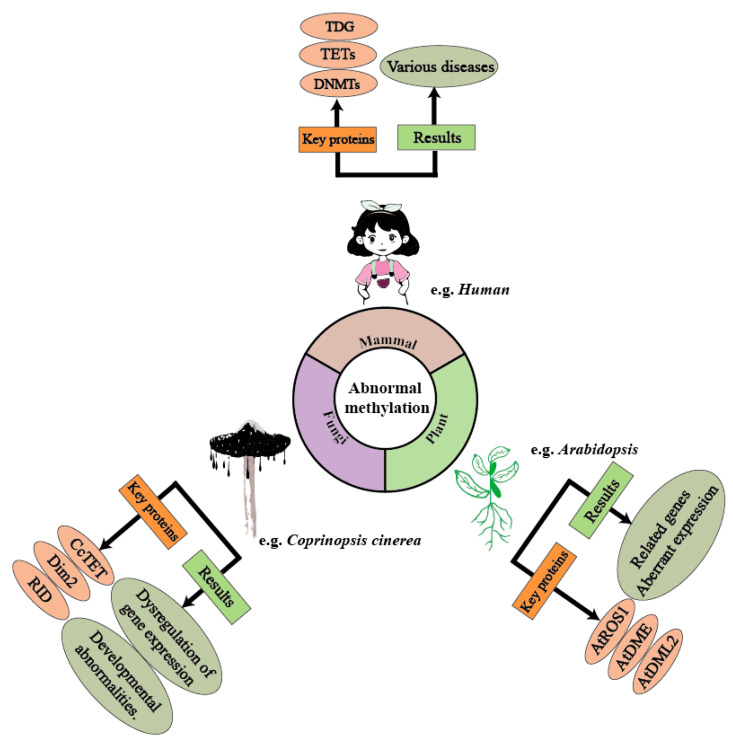
Methylation plays a crucial regulatory role across various biological domains, and abnormal methylation can lead to several negative effects. In humans, abnormal methylation can trigger various diseases through the dysregulation of enzymes such as TETs and the DNMT family. In *Coprinopsis cinerea* (mushroom-forming fungi), the *CcTET* gene is a key factor influencing its methylation levels. Abnormal methylation levels can result in the up-regulation or down-regulation of certain genes, thereby affecting cellular functions. In *Arabidopsis*, genes/proteins such as *ARGOS*, *ADM1*, and *ADM2* influence DNA methylation levels. Under abnormal methylation regulation, *Arabidopsis* exhibits expression dysregulation, leading to developmental abnormalities.

**Figure 2 biology-14-00461-f002:**
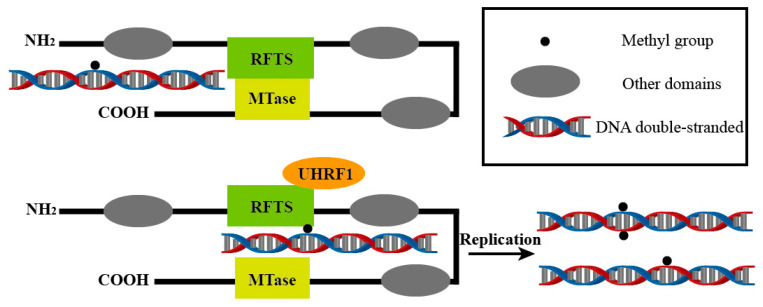
UHRF1 activates the catalytic function of DNMT1 by interacting with the RFTS domain. This allows the existing methylation patterns to be passed on to the next generation during DNA replication and may also generate new methylation sites. This process is central to maintaining DNA methylation and is essential for the stable transmission of epigenetic information.

**Figure 3 biology-14-00461-f003:**
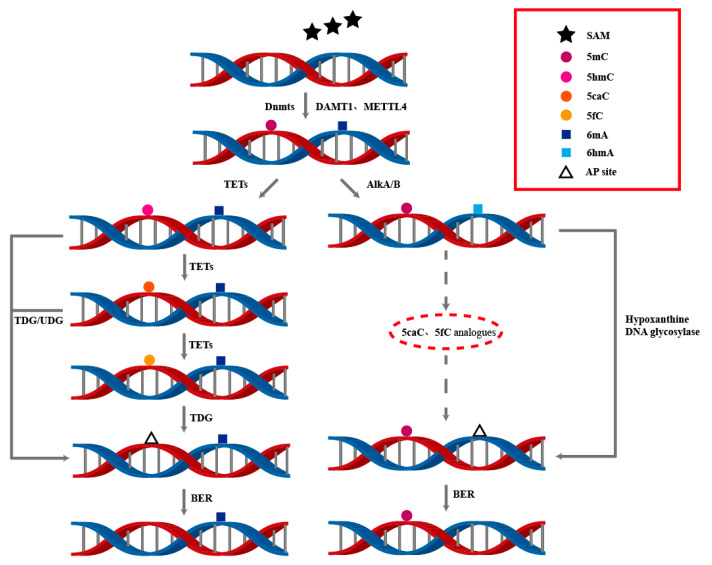
The dynamic processes of DNA 5mC and 6mA modifications involve the interplay of multiple enzymes to achieve the generation and removal of DNA methylation. This is of great significance for understanding DNA modifications and their regulatory mechanisms in epigenetics.

**Figure 4 biology-14-00461-f004:**
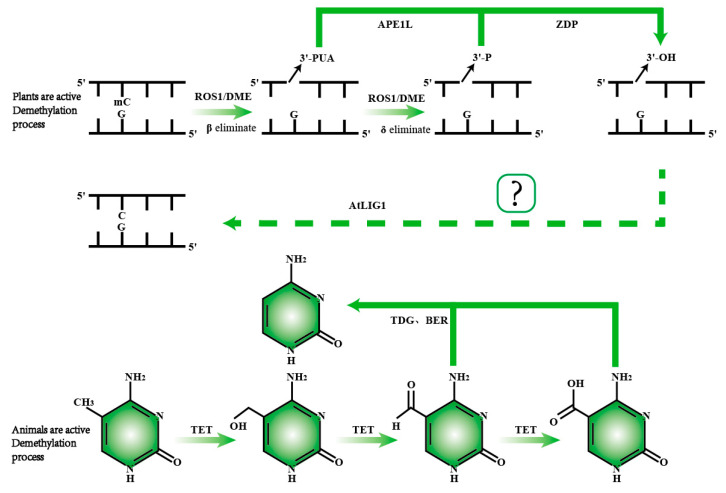
Comparison of active DNA demethylation mechanisms in plants and animals. In plants, ROS1/DME is responsible for recognizing and excising 5mC. Following the removal of 5mC, the APE1L enzyme cleaves the phosphodiester bond, generating a 3′-phosphate end. Subsequently, the ZDP enzyme converts the 3′-phosphate end into a 3′-hydroxyl end. The repair of the DNA strand is then completed by the ALIG1 enzyme, restoring normal base pairing. In animals, active demethylation primarily involves a series of oxidation reactions catalyzed by the TET enzyme, which progressively converts 5mC into 5hmC, 5fC, and 5caC. The TDG enzyme then recognizes 5fC and 5caC, completing the demethylation process through base excision repair.

**Table 1 biology-14-00461-t001:** Comparison of Differences Between ZDP and APE1L.

ZDP	APE1L	Literatures
Dysfunction results in DNA hypermethylation of approximately 1500 endogenous sites, respectively	Dysfunction results in DNA hypermethylation of approximately 3500 endogenous loci, respectively	[85,89]
It is expressed in both vegetative and reproductive tissues	It is mainly expressed in organs	[86]
Effect of ZDP mutation on the TE (transposon) region	The APE1L mutation preferentially leads to DNA hypermethylation of gene regions	[87,88]

## Data Availability

The original contributions presented in this study are included in the article. Further inquiries can be directed to the corresponding authors.

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
