# Peer review of "Cross-Kingdom DNA Methylation Dynamics: Comparative Mechanisms of 5mC/6mA Regulation and Their Implications in Epigenetic Disorders"

_biology, 2025, doi:10.3390/biology14050461_

Round 1
Reviewer 1 Report
Comments and Suggestions for Authors
The manuscript “Cross-Kingdom DNA Methylation Dynamics: Comparative Mechanisms of 5mC/6mA Regulation and Their Implications in Epigenetic Disorders” provides a comprehensive and good overview of the epigenetic roles of DNA methylation across animals, plants, and fungi. This manuscript is interesting and makes a valuable contribution to knowledge of a wide readership in epigenetics and molecular biology. However, I have a few comments to help the authors further improve the clarity, completeness, and applicability of this review, which should not be difficult to address.
Major Comments:
Line 462 Section 6 “Epigenetic Regulatory Network of DNA Methylation in Fungi”: While the fungal methylation section is well-written, it omits key model filamentous fungi such as Neurospora crassa, Trichoderma reesei and Aspergillus niger, which are widely studied in both basic epigenetic research and industrial applications. Adding the relevant findings in these species would make the fungal perspective more representative and informative.
Line 561 Section 7. Conclusions and Prospects: No mention of CRISPR/Cas9-based epigenetic editing in fungi. This review does not include recent advances in CRISPR/dCas9-mediated DNA methylation editing systems applied in filamentous fungi. For example, Li et al. (2021) reported epigenome manipulation via CRISPR in Aspergillus niger (CRISPR/dCas9-mediated epigenetic modification reveals differential regulation of histone acetylation on Aspergillus niger secondary metabolite). These emerging technologies are highly relevant and should be briefly introduced, particularly in the "Prospects" section.
Minor Comments:
Consistency in abbreviation usage:
Please ensure that all abbreviations (e.g., 5mC, DNMT, TET, METTL) are defined upon first use and used consistently throughout the text.
Figure 1 fungal examples too limited:
In Figure 1, the list of "key proteins" under fungi includes only one example. To better represent fungal methylation mechanisms, additional proteins such as Dim-2, RID, and Dnmt5 from Neurospora crassa or fungal methyltransferases from Aspergillus species should be included.
Add the references column to Table 2:
Table 2 summarizes key genes and changes in 6mA Levels in different diseases, which is informative. However, the table currently lacks source references for each listed diseases or changes. Please consider adding a “Reference” column to improve traceability and allow readers to consult the original studies more easily.
Line 485, 486, 491, 535, 550: Latin species names formatting. Some Latin binomials (e.g., Beauveria bassiana, Metarhizium robertsii, M. robertsii, and B. cinerea) are not consistently italicized. Please check the entire manuscript and ensure all species names are properly formatted according to scientific standards.
Author Response
Comment 1: The manuscript “Cross-Kingdom DNA Methylation Dynamics: Comparative Mechanisms of 5mC/6mA Regulation and Their Implications in Epigenetic Disorders” provides a comprehensive and good overview of the epigenetic roles of DNA methylation across animals, plants, and fungi. This manuscript is interesting and makes a valuable contribution to knowledge of a wide readership in epigenetics and molecular biology. However, I have a few comments to help the authors further improve the clarity, completeness, and applicability of this review, which should not be difficult to address.
Response 1: Thanks for your positive comments. We revised all your comments and suggestions.
Comment 2: Line 462 Section 6 “Epigenetic Regulatory Network of DNA Methylation in Fungi”: While the fungal methylation section is well-written, it omits key model filamentous fungi such as Neurospora crassa, Trichoderma reesei and Aspergillus niger, which are widely studied in both basic epigenetic research and industrial applications. Adding the relevant findings in these species would make the fungal perspective more representative and informative.
Response 2: Thanks for your constructive suggestion. We have added more information on these fungi in this section and others. (Line 55-65; Line 456-460; L485-503)
Comment 3: Line 561 Section 7. Conclusions and Prospects: No mention of CRISPR/Cas9-based epigenetic editing in fungi. This review does not include recent advances in CRISPR/dCas9-mediated DNA methylation editing systems applied in filamentous fungi. For example, Li et al. (2021) reported epigenome manipulation via CRISPR in Aspergillus niger (CRISPR/dCas9-mediated epigenetic modification reveals differential regulation of histone acetylation on Aspergillus niger secondary metabolite). These emerging technologies are highly relevant and should be briefly introduced, particularly in the "Prospects" section.
Response 3: Thanks for your constructive suggestion. We introduced the emerging technologies on fungi and cited this literature you mentioned (Line 55-65).
Minor Comments:
Consistency in abbreviation usage:
Comment 4: Please ensure that all abbreviations (e.g., 5mC, DNMT, TET, METTL) are defined upon first use and used consistently throughout the text.
Response 4: Thanks. We have checked all abbreviations throughout the text.
Comment 5: Figure 1 fungal examples too limited:
In Figure 1, the list of "key proteins" under fungi includes only one example. To better represent fungal methylation mechanisms, additional proteins such as Dim-2, RID, and Dnmt5 from Neurospora crassa or fungal methyltransferases from Aspergillus species should be included.
Response 5: Thanks for your constructive suggestion. We revised Figure 1 as your suggestion.
Comment 6: Add the references column to Table 2:
Response 6: Thanks. We added the references column. (Table 1 in this version; L398)
Comment 7: Table 2 summarizes key genes and changes in 6mA Levels in different diseases, which is informative. However, the table currently lacks source references for each listed diseases or changes. Please consider adding a “Reference” column to improve traceability and allow readers to consult the original studies more easily.
Response 7: Thanks for your suggestion. We have modified this table and added references column. (now see Table 1; L398)
Comment 8: Line 485, 486, 491, 535, 550: Latin species names formatting. Some Latin binomials (e.g., Beauveria bassiana, Metarhizium robertsii, M. robertsii, and B. cinerea) are not consistently italicized. Please check the entire manuscript and ensure all species names are properly formatted according to scientific standards.
Response 8: Sorry. We check all the latin species names. And it was confirmed that they are all in italic. (L427-428; L502-514; L596-604)
Reviewer 2 Report
Comments and Suggestions for Authors
This manuscript by Liu et al. compares and contrasts the features of two types of DNA methylation, 5-methyl-cytosine (5mC) and N6-methyladenine (6mA). It also compares and contrasts these modifications between animals, plants, and fungi. Finally, it claims to discuss their implications in epigenetic disorders, but these facts are somewhat dispersed throughout the manuscript, so it is very hard to follow, and they do not cover the role of 5mC in epigenetic disorders, “given the extensive research” covering it elsewhere.
This review is not well-organized, and in trying to cover so many topics about DNA methylation, it makes it very hard to understand what the main idea is. It could compare the two types in more detail, or compare across kingdoms in more detail, or compare their roles in epigenetic disorders in more detail, but in trying to do all three it comes across as a very disorganized list of unrelated facts. Many paragraphs include a lot of data that isn't referenced. For example, in section 3.2 about the demethylation activity of the DNMTs lines 225-229 has somewhat debated data that can not be verified as there are no references.
Also, the wording is sometimes overly complicated, while concepts are often overly simplified, and there are many pointless non-sensical sentences (i.e. line 563 “Despite being forms of methylation, the proportion of 5mC in organisms is significantly higher than that of 6mA.”), giving the writing an AI feel.
I recommend major revisions for this manuscript, including choosing one main focus, removing the parts on epigenetic disease, including more frequent references, and rewriting it in a more appropriate tone.
Author Response
Comment 1: This manuscript by Liu et al. compares and contrasts the features of two types of DNA methylation, 5-methyl-cytosine (5mC) and N6-methyladenine (6mA). It also compares and contrasts these modifications between animals, plants, and fungi. Finally, it claims to discuss their implications in epigenetic disorders, but these facts are somewhat dispersed throughout the manuscript, so it is very hard to follow, and they do not cover the role of 5mC in epigenetic disorders, “given the extensive research” covering it elsewhere.
Response 1: First and foremost, I would like to express my gratitude for your review and constructive suggestions. We have done our best to make this manuscript more accessible and understandable, and to improve its structure. Some redundant contents have been removed as per your suggestion, and some missing contents have also been supplemented.
Comment 2: This review is not well-organized, and in trying to cover so many topics about DNA methylation, it makes it very hard to understand what the main idea is. It could compare the two types in more detail, or compare across kingdoms in more detail, or compare their roles in epigenetic disorders in more detail, but in trying to do all three it comes across as a very disorganized list of unrelated facts. Many paragraphs include a lot of data that isn't referenced. For example, in section 3.2 about the demethylation activity of the DNMTs lines 225-229 has somewhat debated data that can not be verified as there are no references.
Response 2: Thank you for your constructive suggestions. We have added the content on fungal epigenetics based on the suggestion from another reviewer to highlight the details across kingdoms. Furthermore, we have also added the missed but relevant literatures. (L484-504; L517-513; L626-641)
Comment 3: Also, the wording is sometimes overly complicated, while concepts are often overly simplified, and there are many pointless non-sensical sentences (i.e. line 563 “Despite being forms of methylation, the proportion of 5mC in organisms is significantly higher than that of 6mA.”), giving the writing an AI feel.
Response 3: Thank you. We have deleted some irrelevant paragraphs to make the text flow more smoothly and logically.
Comment 4: I recommend major revisions for this manuscript, including choosing one main focus, removing the parts on epigenetic disease, including more frequent references, and rewriting it in a more appropriate tone.
Response 4: Thank you for giving us this opportunity to make corrections. We have made the revisions in accordance with your constructive suggestions above. (L40-46; L80-89; L91-94; L400-403; L654-659)
Round 2
Reviewer 2 Report
Comments and Suggestions for Authors
Authors have addressed my concerns.